# Optimization of Single-Point Incremental Forming of Polymer Sheets through FEM

**DOI:** 10.3390/ma16010451

**Published:** 2023-01-03

**Authors:** Antonio Formisano, Luca Boccarusso, Massimo Durante

**Affiliations:** Department of Chemical, Materials and Production Engineering, University of Naples Federico II, P.le V. Tecchio 80, 80125 Napoli, Italy

**Keywords:** incremental sheet forming, polymers, FEM analysis, toolpath strategy, forming forces, manufacturing time, sustainable manufacturing

## Abstract

Incremental sheet forming represents a relatively new process appointed to form sheets of pure metals, alloys, polymers, and composites for the manufacture of components in fields where customized production in a short time and at a low cost is required. Its most common variant, named single-point incremental forming, is a flexible process using very simple tooling; the sheet is clamped along the edges and a hemispherical-headed tool follows a required path, to deform the sheet locally. In so doing, better formability is reached without any dedicated dies and for low-forming forces, which represent some of the attractive features of this process. Nevertheless, and with special reference to thermoplastic sheets, incremental formed parts suffer from peculiar defects like twisting and wrinkling. In this numerical work, analyses were conducted through a commercial finite element code by varying the toolpath strategy of the incremental forming of polycarbonate sheets. The investigation of some features like the forming forces, the deformation states, the energy levels, and the forming time was carried out, to determine the toolpath strategy able to optimize the incremental forming process of polymer sheets. The results of the numerical analyses highlight a reduction of the forming forces when using toolpaths alternating diagonal up and vertical down steps and, presumably, a reduced risk of failures and defects. Furthermore, these toolpath strategies solutions also have a positive impact on the environment in terms of energy and do not significantly increase the manufacturing time.

## 1. Introduction

The industrial revolution of the last decades demands manufacturing processes with less changeover time and tooling cost; consequently, conventional manufacturing processes might prove ineffective for small batch production and prototypes. In addition, the advances in the use of computers applied to manufacturing have encouraged the development of procedures with higher levels of flexibility, (for instance, those not requiring dedicated dies). In this context, incremental sheet forming (ISF) has begun devoting much greater attention in recent years, due to its flexible and cost-effective nature that enables it to respond to the above-mentioned challenges [1].

ISF is born as an excellent alternative technique to other material forming procedures to deform incrementally flat metal sheets into the preferred complex three-dimensional profile and, to do this, a computer numerically controlled (CNC) generic tool stylus acts on a sheet of material, peripherally clamped along its outer edges [2]; among the different existing ISF process variants, the most basic is known as single-point incremental forming (SPIF), and involves the use of a simple tool and the absence of dies for the superimposition of local deformations in the sheet [3].

The main characteristics of this flexible process are higher formability, low forming forces, reduced lead time, and cost-effectiveness [4], while the applications cover several industrial fields like aerospace, on-site repair of military components, prototypes in automotive, as well as customized products in medical, architecture, etc. [5].

ISF process is widely used for several metals and alloys (also characterized by hard workability) like aluminum, steel, copper, titanium, etc. [6], but it was recently extended to polymers and composites [7].

Starting from the preliminary studies of Le et al. [8] and Franzen et al. [9], highlighting the goodness of the above mentioned process for the manufacture of complex polymer sheet components, the interest in the ISF of thermoplastic polymers has been increasing over the past few years enough to be a valid alternative to conventional technologies, based on heating-shaping-cooling manufacturing routes; the latter ones (i.e., extrusion, molding, casting, and thermoforming) are only economically viable for mass production, because of the high costs tied to the tool’s design and manufacturing [10]. Conversely, one of the main benefits of working polymers by ISF is that the process can still be carried out at room temperature, achieving high levels of material formability [11]. The applications of parts obtained by ISF of these strongly engineered materials range from many fields, in particular aerospace and unmanned aerial vehicles, like racing and commercial cars [12], and customized products in the biomedical sector [13].

Failures and undesired deformation phenomena like twisting and wrinkling interest ISF sheets, including but not limited to the polymer ones [14,15]. Twisting is due to the component of the forming forces tangential to the toolpath, that generates in-plane shear and, consequently, an uncontrolled twist of the sheets with respect to the clamping frame; it is also promoted by higher levels of vertical forming forces when forming polymer sheets, since they are affected by significant indentation that accentuates the phenomenon [16,17]. The twisting was investigated and observed by the authors on axisymmetric components obtained by a unidirectional toolpath both for aluminum alloy [18] and polycarbonate sheets [19]; the twisting angles for the latter ones were very high, compared to the sheet metal ones (about 22° vs. less than 6°), and a dramatic reduction of the phenomenon was possible adopting an alternate toolpath. All the same, severe forming conditions in terms of sliding forces could generate wrinkling, in particular when forming thin thermoplastic sheets, characterized by low mechanical resistance [20].

Measuring and predicting the forming forces during ISF is a major research area since it is an efficient tool for monitoring the quality of the process [21]. Besides, the reduction of the forces acting in the sheet plane represents a way to reduce the risk of failures and defects on the ISF of polymer sheets; in addition, lower sliding forces translate into lower global forming forces with a reduced risk of tool failure, improvement of the formed surfaces’ quality, and the chance to reduce lubricants to lower friction and sticking of material to the tool. Finally, it can also involve energy implications; sustainable manufacturing is a hot topic in the face of global warming and, consequently, the improvement of the industrial processes also goes through the mitigation of their negative impact on the environment in terms of energy [22].

Considering the above, it is evident that it is advisable to perform an optimization strategy for the process under examination; besides, optimization methods were largely considered for enhanced performance in several engineering cases [23,24,25]. The choice of the toolpath, controlled by a part program generated by computer-aided manufacturing (CAM) software, represents a very significant success factor for the considered manufacturing process, because of its relevant effect on different aspects like dimensional accuracy, thickness distribution, processing time, and surface roughness [26]. Consequently, the paper aims to individuate a toolpath strategy useful for optimizing the ISF of polycarbonate through a numerical approach based on Finite Element Method (FEM) simulations; an accurate lecture of FEM results represented an optimization tool in a direct (manufacturing time and energy states) and indirect way (prediction of defectiveness and risks of failures as a function of the forming forces and the energy levels). As it is well known, FEM consists of a numerical technique to find approximate solutions to partial differential equations of a system [27]; it allows producing much more detailed results than experimental investigations, and is often quicker and less expensive. FEM simulations were largely employed to increase understanding of phenomena interesting ISF of polymer sheets. For example, the authors determined stress, strain, and thickness distributions during the process [20], and developed a thermo-mechanical numerical model to investigate the suitability of friction heating, generated by the forming tool rotation, to form polymer sheets during ISF [28]. Moreover, Medina-Sanchez et al. [29] proposed a model to predict axial force in SPIF of thermoplastic sheets, while FEM aided the investigation of the feasibility of an advanced robotized polymer ISF in [30].

Concerning the polycarbonate, it is considered as a “transparency metal”, due to its remarkable mechanical and physicochemical properties (such as light-weight, strength, corrosion resistance, and price, among others) [31]; polycarbonate parts find application in the areas of communications and transport, medical apparatus and instruments, the aerospace environment, etc. [32].

The manufacture of a fixed wall angle cone frustum through SPIF and by setting five different toolpaths, starting from polycarbonate sheets at room temperature, was simulated by a commercial FEM code. Some outputs of the simulations were investigated, like the forming forces, the manufacturing time, different forms of energies, and the stress-strain states, to draw conclusions on how the toolpath strategy influences the process and individuates the best solution in terms of reduced risk of failures and defects, surface quality, and energy implications.

## 2. Materials and Methods

The FEM commercial code LS DYNA was used to simulate the incremental forming process under study; this software is a general-purpose finite element program capable of simulating complex real-world problems, widely used by, among others, the automobile, aerospace, construction, military, manufacturing, and bioengineering industries. The simulations were carried out by considering the equipment and the materials at the disposal of the authors (as well as typical process parameters), while the efficiency of the code was guaranteed by different studies present in the literature [33,34], including some authors’ works; in particular, they considered using this FEM code both for metal (aluminum alloy sheets [18]) and polycarbonate sheets, in the last case for foreseeing the occurrence of wrinkling [35].

The numerical model was constituted by the sheet, a square with a side *L* = 100 mm (equal to the internal area of the clamping frame) and thickness equal to *t* = 1.5 mm, and the hemispherical head of the tool (the part in contact with the sheet; radius *r* = 5 mm).

The components, i.e., conical frusta, presented a major base with radius *R* = 35 mm, height *h* = 20 mm, and a wall angle *α* = 60°. The main characteristics of the equipment and of the components to manufacture are schematized in Figure 1.

Both the sheet and the tool were simulated using shell elements and opportune materials’ models; the main characteristics of the FEM model (properties of the elements and of the materials, boundary, and contact conditions) are reported in Table 1.

In detail, the tool was considered with rigid behaviour, compared to the polycarbonate sheet, and a model for the tooling in forming applications was considered, coupled with the default type of shell elements that gives extremely cost-effective computational solutions. The material’s model used for the sheet was useful for elastoplastic polymers, while the shell elements formulation was capable of simulating the deformable parts in forming problems.

Fixed constraints were assigned to a set of nodes (the peripheral nodes of the sheet) to simulate the action of the clamping frame.

The interaction between the two parts was governed by a contact card: the friction coefficient was set in line with a numerical work present in the literature [36]. However, consider that in experimental tests like the simulated ones the risk of damages is limited by lubricating the sheets with mineral oil for cold forming; in so doing, friction and sticking of material to the tool were reduced [20].

Five different toolpath strategies were considered in this study. The reference toolpath involved the contact of the tool with the sheet on points of a spiral path; with this solution, the tool follows a continuous path along X, Y, and Z axes, outlining the shape of the geometry and avoiding line scarring caused by step downs, typical of a Z-level contouring toolpath [37]. The spiral was described with a vertical distance between two successive spirals equal to *vs* = 1.0 mm; the points along the spiral were angularly spaced around *θ* = 6° from each other. The schematization in Figure 2, limited to one spiral, reports the characteristics of this path; in detail, it reports *vs* and *θ*, while three consecutive points are labelled with A to C.

As anticipated above, four other strategies were considered to link two consecutive points; the first one was a stair path (involving an alternation of horizontal and vertical down steps), while the other three alternated between diagonal up and vertical down steps. For these four cases, the ramp height of the first step, *hr*, was equal to 0 (stair path), 0.5, 1.0, and 1.5 mm. Figure 3 summarizes, in a not-to-scale representation, the planar development of the five toolpath strategies; they are labeled *ref_tp* (reference toolpath) and *hr*0*_tp*, *hr*0.5*_tp*, *hr*1.0*_tp*, and *hr*1.5*_tp*, as a function of *hr*.

The tool motion with a feed rate equal to *v* = 1000 mm/min was assigned by means of the X, Y, and Z displacement laws (a series of interpolation points represented by the cartesian coordinates coupled to the corresponding manufacturing times), created with a Microsoft Excel spreadsheet developed by the authors.

Due to the long toolpaths that characterize the ISF processes, the computational time of the simulations could be very long; in order to reduce it, the mass scaling was used [38].

## 3. Results

This section summarizes the main results from the simulation campaign. They are commented on in the following Discussion section, along with other results.

The forming forces were collected by the most common contact-related output file, RCFORC. It is an ASCII file containing resultant contact forces, written in the global coordinate system, for the slave and master sides of each contact interface. Figure 4 reports the trend of the forming forces vs. time for the two limit cases, i.e., the *ref_tp* (Figure 4a) and the *hr*1.5*_tp* (Figure 4b). Note that *F_X_*, *F_Y_*, and *F_Z_* were directly obtained from the RCFORC file, while the module of the force acting in the sheet plane, *F_XY_*, was the combination of *F_X_* and *F_Y_*:(1)$$F_{XY}=\sqrt{F_{X}^{2}+F_{Y}^{2}}\;;$$

Moreover, these were the forces that the slave (the sheet) transmits to the master (the forming tool), with the respect of the coordinate system in Figure 1.

To appreciate quantitatively the influence of the toolpath strategy, Figure 5a,b report the trends of *F_Z_* and *F_XY_* for the five different toolpaths. Note that, in these figures, and in contrast to Figure 4, the time on the abscissa axis is expressed in percentage terms, with respect to the processing time of each case, for a simpler comparison among the force trends. In fact, the forming times are not the same; concerning this, Figure 6 reports the forming time vs. the toolpath strategy.

Finally, Figure 7 reports three forms of energy vs. the toolpath strategy. In detail, they are the total energy, *E_t_*, the sliding energy, *E_s_*, and the internal energy, *E_i_*. They were collected by the ASCII file named GLSTAT.

## 4. Discussion

From Figure 4, the *ref_tp* case shows the typical trend of the forming forces for a SPIF process for the manufacture of a cone frustum with a spiral toolpath [20,39]. This trend was also observed for the *hr0_tp* case but it was not represented in the figure, to allow a better result readability. *F_Z_* gradually increases with time, until it stabilizes. The oscillations are due to a little variability of the stiffness of the sheet, since during a spiral, the distance of the tool from the frame is different (minimum along X and Y axes, maximum along the diagonals) and, with it, the mechanical reaction of the sheet. *F_X_*, *F_Y_*, and *F_XY_* increase with time until reaching the steady-state condition with the typical sinusoidal trend of *F_X_* and *F_Y_*, while *F_XY_* presents a trend similar to *F_Z_*; on the contrary, the *hr*1.5*_tp* case, as well as the *hr*1.0*_tp* and *hr*0.5*_tp* strategies (the last two not reported in the figure), show an atypical and irregular trend of the forces.

Figure 5 highlights that *ref_tp* and *hr*0*_tp* strategies determine similar forming forces, slightly higher for the first one; this last sentence is justified by the fact that the *ref_tp* strategy involves a continuous vertical down movement of the tool (and then, the most severe contact conditions), which is different from all the other strategies. For both *ref_tp* and *hr*0*_tp* cases, the typical trend of the forces is due to the continuous tool/sheet contact during all the process; this is obvious for *ref_tp* but is also true for the *hr*0*_tp* case too, due to the elastic springback that guarantees the contact between the tool and sheet, also during the horizontal steps of the toolpath [40]. Starting from the *hr*0.5*_tp* strategy, both *F_Z_* and *F_XY_* decrease significantly; in addition, their trends are completely irregular. *F_XY_* even tends to zero for the last two strategies; this is representative of an almost non-contact between the tool and the sheet on the top of the ramp heights of the toolpath (see the lower *F_Z_* values), because of *hr* values being similar to the entity of the elastic springback.

The differences in terms of forces are reflected in different strain states too. Concerning this, Figure 8 reports the maximum shear strain for two consecutive forming steps related to the two extreme cases (*ref_tp* and *hr*1.5*_tp* strategies), and for about the same percentage manufacturing time (equal to about 25%).

From the figure, it is possible to note that the *ref_tp* strategy (Figure 8a) determines strain accumulation on a large area of the sheet and its distribution is asymmetric, following the advancement of the tool (whose current position is indicated with an arrow); these phenomena are not so noticeable for the *hr*1.5*_tp* strategy (Figure 8b), and this translates into a more localized deformation and reduced distortion of the shells subject to the forming action of the tool.

In light of the results in terms of forming forces (see Figure 5), and considering that higher and more regular plane forces determine a combination of continued strain accumulation and asymmetric strain levels (and, consequently, higher probability of twisting occurrence) [41,42], it can be assumed that the twisting phenomenon can be mitigated by using a toolpath strategy starting from *hr*0.5*_tp*. In addition, the numerical model used in this work does not include wrinkle instability criteria and it is not capable of accurately predicting the occurrence of wrinkling; but, despite this, the results of the simulations allows for the assumption that the occurrence of this defect is less likely to start from the above reported toolpath strategy.

Concerning the forming time (see Figure 6), it increases, passing from the *ref_tp* to *hr*1.5*_tp* case. This is a direct consequence of the increase of length of the toolpath; the first strategy represents the shortest one, to cover with discrete points a spiral, while the other ones gradually diverge from this condition.

From the histograms of the energies (see Figure 7), the total one decreases, passing from the *ref_tp* to *hr*1.5*_tp* case; however, the last three cases show very varied *E_t_* values. By observing the values of *E_s_*, they follow the trend of the total one, and this is in accordance with the observations on the plane forces. The values of *E_i_*, together with the considerations on the sliding energies, portend an increasing work done in permanent deformation, as well as a different way of deforming (from predominant distortion to compression of the sheet); the last observations are in line with the ones from the considerations on the shear strain states in Figure 8.

## 5. Conclusions

The present work follows a numerical approach for the simulation of the incremental forming of polycarbonate sheets; a commercial FEM code was used to simulate the process for the manufacture of a fixed wall angle cone frusta by varying the toolpath strategy, aiming to individuate solutions capable of reducing the forming forces, and with them the risk of failures and defects, and the spending of energy.

The analysis of the results highlights that the toolpaths alternating diagonal up and vertical down steps reduce both vertical and plane forming forces, compared to the reference toolpath (for which, they reach about 450 N and 350 N) and, with them, the probability of occurrence of twisting and wrinkling. In addition, these solutions also guarantee energy savings and reduce the rate of distortion energy, without affecting too much on the manufacturing time.

Considering all the results of the numerical campaign, it can be argued that one of the toolpath strategies with a positive ramp height guarantees low force and energy levels; the *hr*1.0*_tp* case (the solution with a ramp height of 1.0 mm) can be considered the best choice, since for the same total energy (0.51 MJ), guarantees lower sliding energy (linked to the elements’ distortion) compared to *hr*0.5*_tp* (0.09 MJ against 0.21 MJ), and lower manufacturing time compared to *hr*1.5*_tp* (about 10 s less).

Future works could aim to extend the numerical analyses; for example, not only the toolpath but also the shape of the forming tool could be varied. In addition, an experimental campaign reflecting the numerical tests could be carried out to investigate features, like the twisting and the surface roughness, not observable by FEM.

## Figures and Tables

**Figure 1 materials-16-00451-f001:**
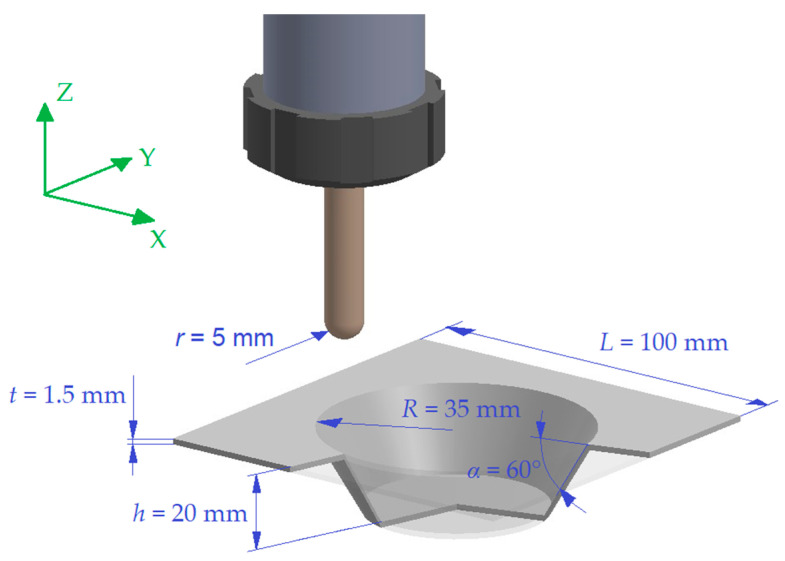
Geometrical features of the equipment and of the cone frustum.

**Figure 2 materials-16-00451-f002:**
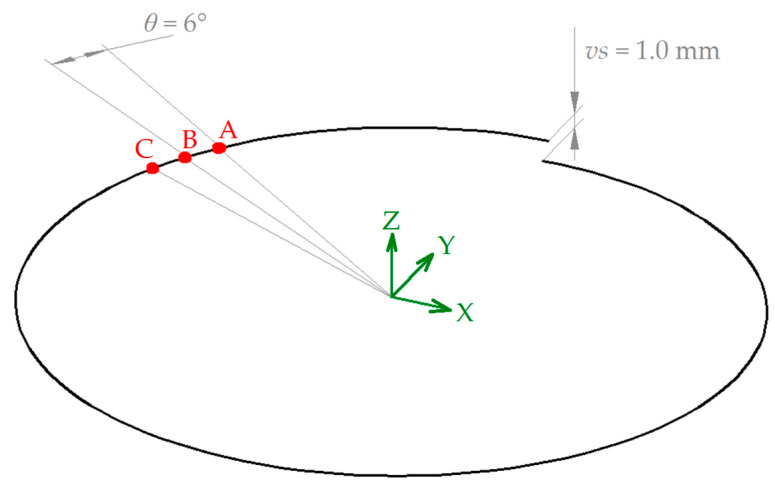
Features and details of the spiral path.

**Figure 3 materials-16-00451-f003:**
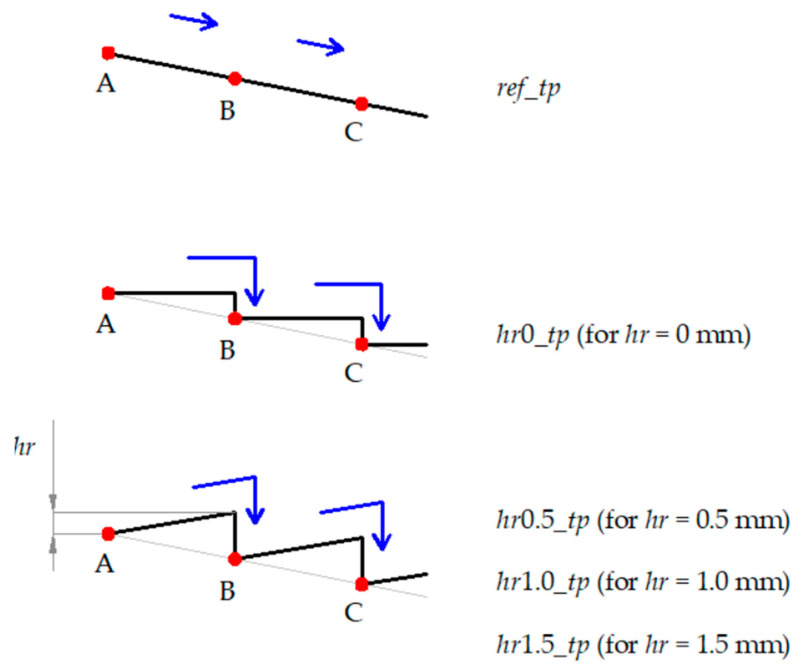
Toolpath strategies.

**Figure 4 materials-16-00451-f004:**
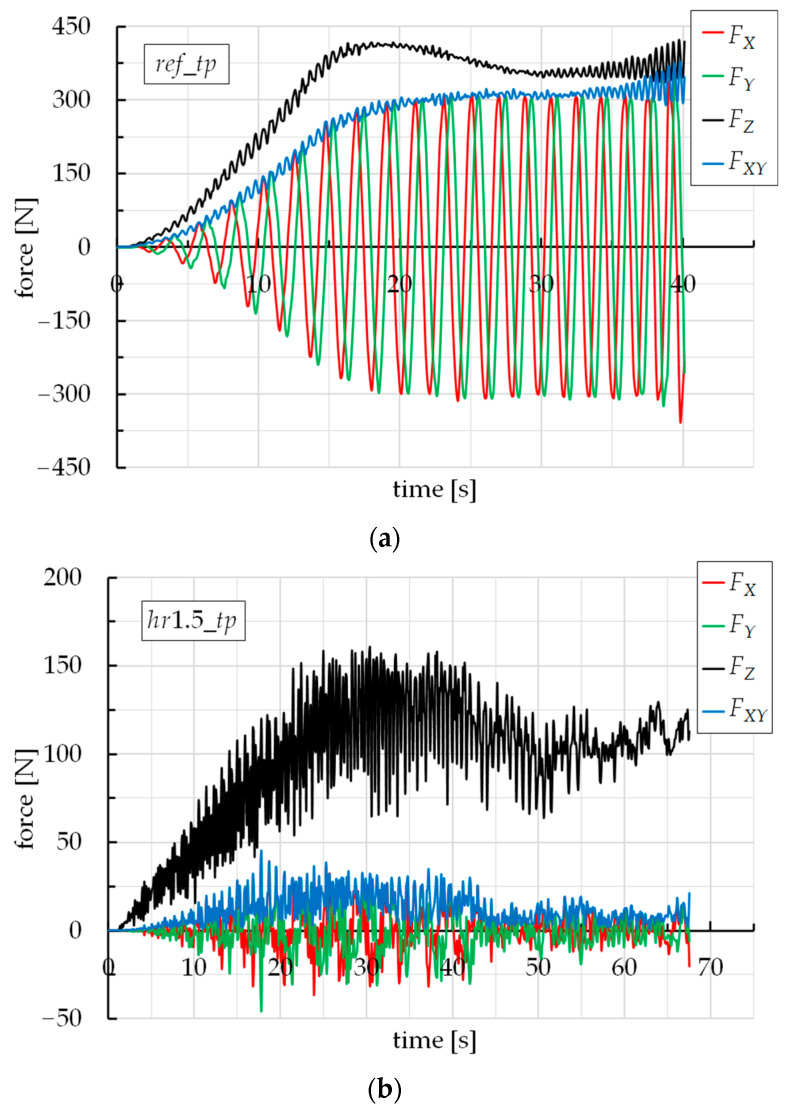
Trends of the forming forces vs. time for (**a**) *ref_tp* and (**b**) *hr*1.5*_tp*.

**Figure 5 materials-16-00451-f005:**
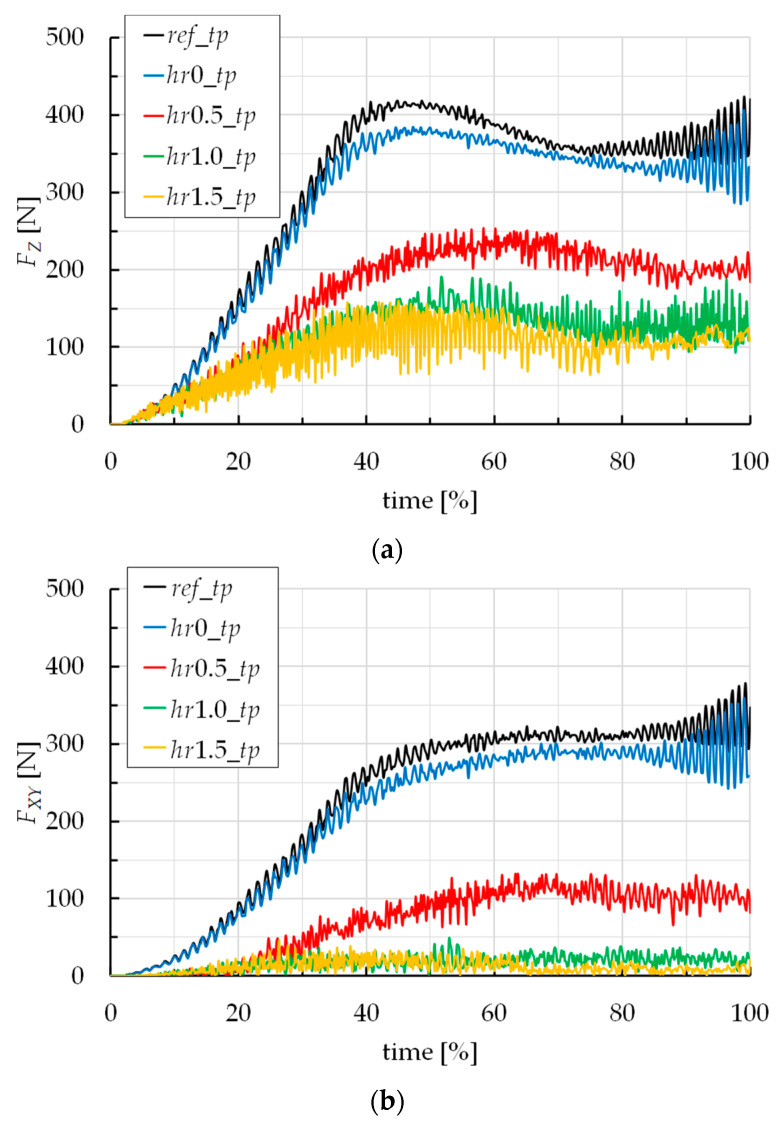
Trends of (**a**) *F_Z_* and (**b**) *F_XY_* vs. time by varying the toolpath strategy.

**Figure 6 materials-16-00451-f006:**
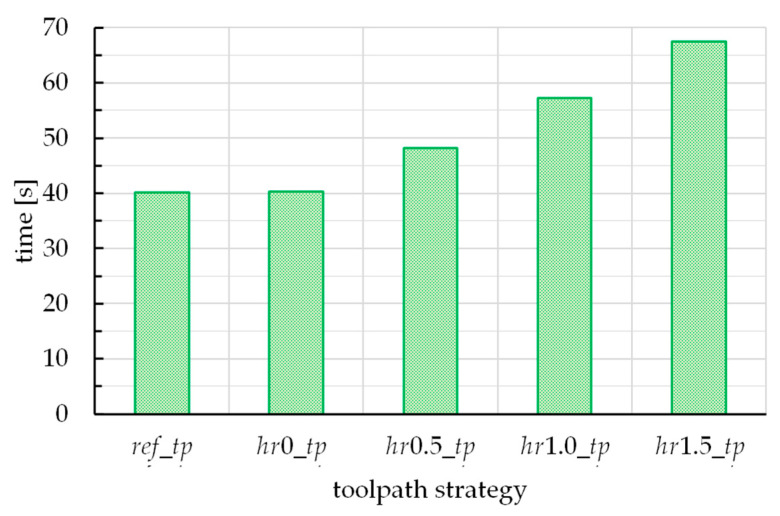
Forming time vs. the toolpath strategy.

**Figure 7 materials-16-00451-f007:**
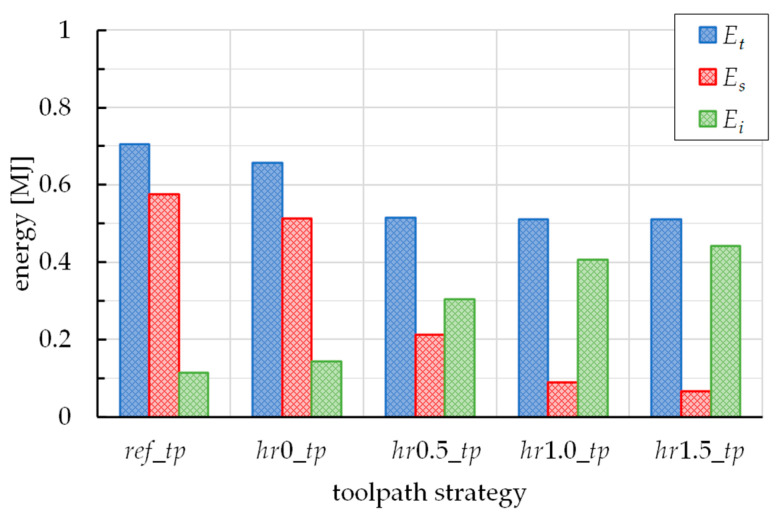
Total, sliding and internal energy vs. the toolpath strategy.

**Figure 8 materials-16-00451-f008:**
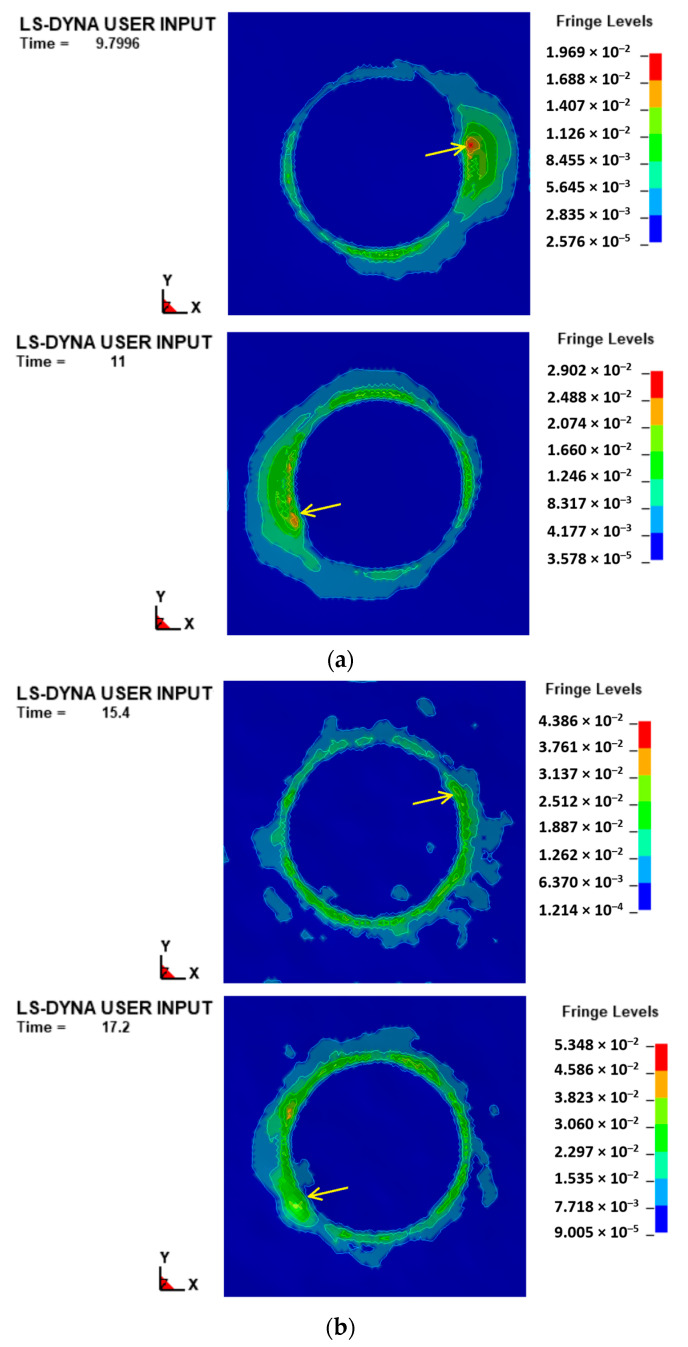
Max shear strain distribution for (**a**) *ref_tp* and (**b**) *hr*1.5*_tp* strategy.

**Table 1 materials-16-00451-t001:** Main properties of the FEM parts.

Part	Sheet	Tool
Element type	Fully integrated shells	Belytschko-Tsay shells
Integration points on the surface	4	1
Integration points through the thickness	5	2
Number of elements	5200	670
Mean dimension of the elements	1.41	0.2
Material model	MAT PLASTICITY POLYMER	MAT RIGID
Density [g/cm^3^]	1.2	7.85
Young’s modulus [GPa]	2.3	210
Poisson’s ratio [-]	0.3	0.3
Yield stress [MPa]	60	-
Ultimate elongation [%]	110	-
Boundary conditions	BOUNDARY SPC: Rotational and translational constraint of the peripheral nodes	BOUNDARY PRESCRIBED MOTION: X, Y and Z translations for rigid part
Contact conditions	SURFACE TO SURFACE: friction coefficient equal to 0.33	

## Data Availability

Not applicable.

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
