# Peer review of "Optimization of Single-Point Incremental Forming of Polymer Sheets through FEM"

_materials, 2023, doi:10.3390/ma16010451_

Round 1

Reviewer 1 Report

The paper is suitable for publication after the authors have responded to some of the following comments:
1. Please demonstrate the novelty of the present work at the introduction stage.
2. please refer to papers presenting optimization problems (DOI:) 10.12913/22998624/61931, 10.1016/j.ifacol.2017.08.2287, 10.1016/j.enbuild.2017.09.065.
3 Please enlarge the figures within Figure 8.
4. Conclusions should relate more to quantitative evaluation.

Reviewer 2 Report

It is a pleasure to review a paper that presents a research regarding such a topic which, in my opinion, is quite rarely published.

Chapters 1 contain a good introduction/knowledge and even a short “state of the art”regarding the ISF in general and the polycarbonate forming by SPIF in particular. This short review prepares the readers for what follows forward in the paper. The motivation of the research is done and the main conclusions aim to be drawn from this research is presented.

Chapter 2 present a description of the processes, the method, the part, material and so on, thus is a satisfactory chapter. Even so, two questions and a recommendation is made:

- Question 1: How is was chosen the feed rate of 1000mm/min and the step depth of 1mm ? - comment your choice (based on the literature or you have other arguments ?)

- Question 2: Very interesting toolpaths were chosen to simulate the model, but how their was obtained ? By CAM software ? If yes, which software and which “milling” strategy ? If not, please explain how the toolpaths were obtained !

- Recommendation: It is recommended to describe more the FEM model, the mesh startegy and I thing it is very important for many researchers to find how the tool trajectory were implemented in LS Dyna (probably as succession of interpolation points, but what were the steps folowed in LS Dyna)

Chapter 3 present the results in a clear manner. Is good that you highlighted that the toolpath time does not increase significantly for each strategy.

Chapter 4 – the result are in detail commented.

Conclusions are well claimed. There it is highlighted that the diagonal trajectory reduce the main defect for SPIF of polymer sheets which, in my opinion is the main claim of the paper. Future work and research topics are also proposed.

Another text remark - lines 99, 203 and 260 – maybe “cone frustum” instead of “cone frusta”.

Overall, the manuscript is a quite good paper, well written and easy to understand.

Reviewer 3 Report

The manucript can be accepetd after the minor revisions:

1. Add more keywords for the readers

2. Mention the research gaps properly.

3. Make sure that all the abbreviations are defined at the place where they are used for first time.

4. Add some new literature to enhance the introduction part.

5. Conclusion should address that whether the aim was achieved or not.

6. Language is fine, minor spell check is required.

7. State the motivation of the study in the introduction part.

Reviewer 4 Report

Dear Authors

The authors have attempted Optimization of Single-Point Incremental Forming of Polymer Sheets through FEM. The investigation carried out with forming forces, the deformation states, the energy levels, and the forming time to optimize the incremental forming process of polymer sheets. The manuscript is well written with adequate design and methods and the topics could more interesting for researchers.

The manuscript needs revision based on following comments and suggestions.

1.      Include the quantitative value of key finding in the abstract.

2.      Lack of comprehensive literature support for relevant FEM modeling of polymer; hence include some more recent relevant literature papers in the introduction sections.

3.      Authors should include chemical compositions of Polycarbonate sheets.

4.      Clearly state the boundary conditions of models.

5.      The manuscript has many typo errors. Author need to revise the entire manuscript carefully.

Ex Line No: 120 “eight instead of height”

6.      In result and discussion, include the scientific finding and justification with relevant literature.

7.      Experimental validations are missing. Authors should consider for validations of text results.
